# Comparison of the results of in-person and mobile phone surveys for a health facility assessment in Tajikistan: A validation study protocol

Rachel Neill[1‡]*, Pablo Amor Fernandez[1‡], Ruchika Bhatia[2], Jigyasa Sharma[2], Kathryn Andrews[2], Sven Neelsen[2], Etoile Pinder[3], Marifat Abdullaev[4], Firuza Safarova[5], Mutriba Latypova[2], Mirja Channa Sjoblom[2], Tashrik Ahmed[1], Michael A. Peters[1], Ashley Sheffel[1], Tawab Hashemi[1], Peter M. Hansen[1], Ghafur Muhsinzoda[6], Gil Shapira[2]

**1** The Global Financing Facility for Women, Children, and Adolescents, Washington, District of Columbia, United States of America, **2** The World Bank, Washington, District of Columbia, United States of America, **3** Sanigest Internacional, Panamá, Panama, **4** Department of Medicine, Tajik National University, Dushanbe, The Republic of Tajikistan, **5** Consultant, Dushanbe, The Republic of Tajikistan, **6** Ministry of Health and Social Protection of the Population of the Republic of Tajikistan, Dushanbe, The Republic of Tajikistan

‡ Co-first authors contributed equally to this work.
* rneill@worldbank.org

## Abstract

Health facility assessments provide important data to measure the quality of health services delivered to populations. These assessments are comprehensive, resource intensive, and periodic to inform medium- to-longer-term policies. However, in absence of other reliable data sources, country decision makers often rely on outdated data to address service delivery challenges that change more frequently. High-frequency phone surveys are a potential option to improve the efficiency and timeliness of collecting time-sensitive service delivery indicators in-between comprehensive in-person assessments. The objectives of this study are to assess the reliability, concurrent criterion validity, and non-response rates in a rapid phone-based health facility assessment developed by the Global Financing Facility's FASTR initiative compared to a comprehensive in-person health facility assessment developed by the World Bank's Service Delivery Indicators Health Program. The in-person survey and corresponding in-person item verification will serve as the gold standard. Both surveys will be administered to an identical sample of 500 health facilities in Tajikistan using the same data collection entity. To assess reliability, percent agreement, Cohens Kappa, and prevalence and bias adjusted Kappa will be calculated. To assess concurrent criterion validity, sensitivity and specificity will be calculated, with a cut-off of .7 used for adequate validity. The study will further compare response rates and drop-out rates of both surveys using simple t-tests and balance tests to identify if the

**Data availability statement:** Our intention is to make the deidentified research data publicly available when the study is completed and published.

**Funding:** The authors received no specific source of funding or specific grant or award for this research. This research is being conducted as a research activity alongside funded health survey activities occurring under the Tajikistan Millati Solim (Healthy Nation) project, which is funded by the World Bank Group and the Global Financing Facility for Women, Children, and Adolescents.

**Competing interests:** The authors have declared that no competing interests exist.

characteristics of the phone-based and in-person survey samples are similar after accounting for any differences in survey response rates. The results of this study will provide important insights into the reliability and validity of phone-based data collection approaches for health facility assessments. This is critical as Ministries of Health seek to establish and sustain more continuous data collection, analysis, and use of health facility-level data to complement periodic in-person assessments to improve the quality of services provided to their populations.

## Introduction

Large-scale, in-person health facility assessments (HFAs) offer detailed data on health systems functioning and quality of care, making them indispensable for evidence-informed decision-making and health policy development. Several holistic health facility assessments exist – including the Harmonized Health Facility Assessment (HHFA) [1], the Service Availability and Readiness Assessment (SARA) [2], the Service Provision Assessment (SPA) [3], and the Service Delivery Indicators survey (SDI) [4] – that paint a detailed picture of health facility readiness and service delivery. These surveys are important complements to administrative data in in low- and middle-income countries (LMICs) by providing detailed information on the supply-side of the health system including the availability of key health systems inputs, the functioning of health facility management, providers' clinical knowledge, and patients' satisfaction with care.

Large-scale in-person HFAs are, by design, meant to be comprehensive in their assessment, requiring considerable time, financial and human resources. They are conducted periodically, often every five to seven years, to inform medium-to-longer-term policies and planning. However, in absence of other frontline service delivery data sources that are reliable and comprehensive, country decision makers are often forced to rely on HFA data that is often outdated, with little ability to diagnose and address supply-side service delivery challenges that may be changing more frequently.

In an ideal world, comprehensive in-person HFAs would be complemented with both high-quality administrative data and high-frequency assessments. Large scale, comprehensive in-person HFAs offer a detailed understanding of the health system and are well suited to capture systemic changes that typically unfold over longer periods. Meanwhile, data gathered from high-quality administrative sources and high-frequency assessments can provide timely insights into key indicators expected to change more frequently. This is particularly important in fragile and conflict-affected contexts and during shocks (e.g., epidemics or natural disasters) when frequent contextual changes make timely adaptation and response especially critical. For example, immediate disruptions in human resources or medical supplies – whether due to sudden crises or chronic health systems challenges – can be more swiftly managed using higher frequency, longitudinal data. More broadly, higher frequency and more targeted assessments could provide regular insight into service readiness indicators

that are time variant such as essential medicine availability, while the in-person observation by skilled enumerators found in in-person HFAs is critical for detailed data on provider competencies. Thus, the integration of comprehensive HFAs with ongoing administrative data analysis and high-frequency assessments have the potential to enable a more responsive and resilient health care system as well as improve countries' ability to monitor quality of care and strengthen health systems over time.

Phone surveys are one potential approach to enable high-frequency HFAs. Previous research indicates that phone-based data collection is an affordable approach to collecting both qualitative and quantitative data [5]. During the COVID-19 pandemic, more frequent, locally adapted phone survey methodologies were utilized to monitor essential health service use and the impact of the pandemic on health facility readiness to deliver services [6,7]. This demonstrated that phone-based survey approaches can provide frequent and cost-effective results in LMIC settings [6].

Growing evidence indicates that interview administered phone surveys of households [8–11] and health workers [12–14] can have utility and validity; however, there is less evidence that phone-based health facility assessments can capture comprehensive details on health facility functioning with limited bias. Two studies offer promising results. In Malawi, Pattnaik et al. conducted direct observations and register reviews at facilities to assess select aspects of phone survey validity related to family planning indicators [14]. The study found that most health facility indicators collected by phone were above their predetermined 70% sensitivity threshold to assess validity compared to in-person records and inspections and determined that phone data collection was both feasible and highly cost-effective[14]. In Kenya, Ashigbie et al. assessed the validity of phone-assessed medicine availability indicators at health facilities compared to in-person visits. The study found strong health facility-level agreement on medicine availability between the phone survey and subsequent in-person assessment ([kappa = 0.90; confidence interval (CI) 0.88–0.92]); further, the phone survey was conducted with a unit cost of $19.73 compared to $186.20 for the in-person assessment [15]. Both studies concluded that phone-based surveys were valid and cost-effective compared to existing gold standards of either in-person data collection or inspection visits for a limited set of indicators [14,15].

Despite promising initial results, a key concern for phone-based surveys is minimizing respondent fatigue by keeping the survey administration short. Like in-person surveys, phone-based interviews that take a longer period are more likely to lead to respondent fatigue, which can give rise to threats to validity. For example, recent household survey research quantified that a 15-minute addition in questionnaire administration time resulted in 8-to-17% less detailed response over the phone [16]. Similarly, Abate et al. (2023) randomly assigned household survey participants to in-person or phone-based survey administration for a household nutrition consumption survey [17]. They found that average per-capita consumption reported by respondents was 23% lower and the estimated poverty headcount was twice as high in the phone-based survey compared to the in-person survey due to underreporting of food items consumed, which the authors attributed to respondent fatigue [17]. In contrast, Torrisi et al. (2024) ran a similar experiment in Malawi, randomly allocating participants to 10-, 20-, or 30-minute interviews on parental survival, finding that non-response and data quality was similar but that longer interviews were more reliable at reporting the maternal age of death [18]. These findings emphasize the importance of more research on phone survey reliability and validity and how survey duration recommendations may need to vary by topic and content.

An additional limitation of phone-based data collection is that the information provided by respondents is generally self-reported and cannot be directly verified by enumeration using observation. This poses specific validity questions in health facility surveys which rely on in-person verification of medical equipment, drugs, health workers, registry logs, and physical infrastructure. Moreover, more complex indicators cannot be effectively administered over the phone. For example, while a phone-based survey might approximate in-person measurement for basic infrastructure availability or the number of staff assigned to a health facility, it is likely unsuitable for assessing more intricate constructs like provider performance and adherence to clinical protocols.

Finally, phone-based data collection is likely optimally conducted with a single respondent, given expected operational challenges with 'passing the phone' around the facility and/or feasibility of enumerating and then reaching multiple respondents via phone. In contrast, most large-scale, comprehensive HFAs rely on multiple respondents, often starting with the health facility manager but going on to include respondents across the health facility (e.g., service providers, pharmacist, financial administrator/accountant, etc.) in the course of a single visit. This means that the phone-based survey likely requires a careful selection of the key questions or domains that a single respondent is best suited to answer.

Despite the rise in phone-based surveys, promising early validation results, and these known limitations, there has not been a validation study of a nationally representative, phone-based HFA to our knowledge. This is a critical gap in the evidence – can phone-based surveys measure key constructs that are commonly measured via in-person HFAs? And what bias, if any, is introduced? This study will fill that gap by conducting a head-to-head comparison of a nationally representative, comprehensive, in-person HFA and a nationally representative, rapid, phone-based HFA. Our findings will have important implications on the feasibility of adopting phone-based surveys as a key complement to comprehensive HFAs to improve the timeliness and responsiveness of health systems data to better align with real-world decision-making needs.

## Study objectives and contributions

This study has three primary objectives. The first objective is to assess the inter-method reliability of the rapid phone-based HFA compared to the comprehensive in-person HFA. This objective focuses on isolating the survey mode effect – the differences in results due to the phone and in-person administration – and will determine the extent to which phone data collection is a reliable alternative mode for conducting HFAs for a subset of key PHC service delivery indicators. To minimize bias, this objective will include only the survey questions that are identical in both the rapid phone-based HFA and the comprehensive in-person HFA. By keeping questions the same and varying only the administration mode, we can effectively isolate and examine the mode effect.

The second objective is to assess the concurrent criterion validity of indicators in the rapid phone-based HFA compared to indicators in the comprehensive in-person HFA. As discussed earlier, phone surveys require questions that are shorter, simpler, and less cognitively demanding. While this design is crucial for phone-based assessments, it can lead to issues such as questions being interpreted differently by respondents or lack of clarity or confusion on specific terms in the asked question that may affect data quality. This objective therefore focuses on evaluating the extent to which the rapid phone-based HFA can accurately measure the same underlying indicators as that in a comprehensive in-person HFA via a simplified questionnaire deemed suitable for phone-based administration.

The third objective is to compare the survey response rates and dropout rates of the rapid phone-based HFA to those of the comprehensive in-person HFA. Survey response rates are crucial for interpreting results and identifying potential biases. This is especially important for phone surveys where respondents can more easily drop out or refuse to participate in the survey. For instance, unlike for in-person surveys wherein respondents might feel more obligated to participate in a survey due to the physical presence of the interviewer, the respondents might find it easier to not answer a phone call or hang up. Additionally, phone surveys may also have higher non-response rates due to poor connectivity or time constraints faced by respondents.

Taken together, these objectives will demonstrate whether the rapid phone-based HFA can approximate the results of the comprehensive in-person HFA for a select subset of facility-level indicators. As phone-based data collection in LMICs is a rapidly evolving field with growing interest following the COVID-19 pandemic, publication of this protocol aims to inform the broader scientific community of this undertaking and improve the transparency of our subsequent analysis against the study design articulated below.

## Materials and methods

### Study setting

This study will take place in Tajikistan, a landlocked, and mountainous country in Central Asia. Tajikistan has low population density with most urban areas concentrated in the eastern part of the country [19]. Over 70% of the country's 10.1 million people live in rural areas [19,20]. Health care is overwhelmingly provided in the public sector. The primary healthcare services subject to this validation study are delivered through a combination of rural health centers (RHC) and associated health houses (HH) in rural areas, and by district and city health centers' family medicine departments in urban settings. Primary care provision suffers from dilapidated infrastructure, shortages of equipment, and a low ratio of doctors to the total population, particularly in rural areas [21]. The low ratio of doctors to the population is particularly challenging given Tajikistan's low population density and inaccessibility in mountainous rural areas. To strengthen Tajikistan's health service delivery system and improve population health outcomes, it is critical that health facilities can provide quality services to patients. This includes ensuring that health workers are well trained and capacitated, addressing overcrowding of health facilities, maintaining adequate supplies and drugs, and having an effective referral system in place.

The World Bank (WB) and the Global Financing Facility for Women, Children, and Adolescents (GFF) are supporting the efforts of the Ministry of Health and Social Protection of the Republic of Tajikistan to improve the quality of PHC service delivery. As part of this initiative, they are conducting a series of surveys to assess the quality of PHC in the country. Given Tajikistan's lack of comprehensive data on PHC service delivery, the WB and GFF are supporting the implementation of a comprehensive in-person HFA to get a better understanding of the country's PHC system, identify its strengths and weaknesses, and provide comprehensive data to guide policy decisions. In addition to the comprehensive in-person HFA, the World Bank and GFF are also implementing a phone-based HFA to be conducted every 6–12 months to monitor intervention implementation, inform course correction, and identify new challenges or shocks that emerge over time. Conducting comprehensive in-person HFA and rapid phone-based HFA concurrently in the same setting therefore has offered a unique opportunity to conduct a head-to-head comparison of the results from the two surveys.

### Ethical approval

This study was approved by the Government of the Republic of Tajikistan Ministry of Health and Social Protection Biomedical Ethics Committee, with the Order No. 60 and number N2148 on May 23rd, 2024.

### Survey instruments

The two surveys that will be conducted are: (1) the GFF's Frequent Assessment and Health Systems Tools for Resilience (FASTR) Initiative's rapid-cycle health facility phone survey which is designed for interview-administered phone-based data collection at the PHC level [22] and which will serve as the rapid phone-based HFA and (2) the World Bank's Service Delivery Indicators (SDI) health survey which will serve as the comprehensive in-person HFA [4]. The SDI is a comprehensive survey platform that also includes provider and patient questionnaires; however, for the purpose of this study, we are comparing only a sub-set of SDI's health facility questionnaire to a sub-set of FASTR's health facility questionnaire.

The FASTR rapid-cycle health facility phone survey (subsequently referred to as the rapid phone-based HFA) is a PHC assessment tool that aims to monitor service availability, readiness, and functioning of PHC facilities over time with an emphasis on reproductive, maternal, newborn, child, and adolescent health and nutrition services [22]. The tool's measurement approach is grounded in the WHO and UNICEF's Primary Health Care Measurement Framework and Indicators [23]. The questionnaire was drawn from existing comprehensive HFAs [1–3] as well as health facility phone survey experience during the COVID-19 pandemic [7,24]. The survey is administered to a nationally or sub-nationally representative panel sample of PHC facilities over four quarterly data collection contacts per year. Quarterly data collection enables both the tracking of presumed higher-variability indicators over time (e.g., medicine stock, utility functioning) and the distribution

of annual indicators across four survey rounds to shorten survey administration and reduce respondent fatigue. The phone survey also includes questions that capture respondent perceptions, such those related to challenges faced by the facility. These perception-based questions are a useful way to receive timely feedback from facilities about challenges that they face but are outside the scope of this validation study. As of August 2024, the tool has been implemented in Burkina Faso, Vietnam, Senegal, Bangladesh, Tajikistan, and Madagascar.

The SDI (subsequently referred to as the comprehensive in-person HFA) health survey program began in 2008 and has been implemented in over a dozen countries across the globe [25]. The survey was revamped in 2019 to better align with the recent developments in the literature on measuring quality of PHC service delivery[25,26]. Building on the frameworks for high quality health systems, the survey now assesses processes of care and patient outcomes along with structural inputs to provide a comprehensive picture of PHC service delivery in a country. The SDI survey includes three questionnaires: health facility questionnaire, health care provider questionnaire, and outpatient exit interview. As indicated above, only the health facility questionnaire will be included in the validation study as this is the portion of the survey that aligns with FASTR's health facility questionnaire. The SDI comprehensive in-person HFA, and its corresponding in-person verification of the availability and functionality of structural inputs, will be considered the gold standard for the purpose of this study.

## Questionnaire adaptation

The two tools were adapted to Tajikistan's context via several steps. The initial step involved modifying response options, revising question framing, and adding or removing questions to align with the country's PHC needs and priorities. This included incorporating details such as locally available infrastructure, the country's health workforce structure, clinical guide-lines for diagnosis and treatment of common conditions, and the expected list of tracer equipment, medical supplies, and drugs available at different types of health facilities. Additional questions (including entirely new sections in the question-naires) were developed with input from subject matter experts from WB, GFF, and Tajikistan health systems stakeholders.

This preliminary contextualized questionnaire was shared with a joint working group comprising of local experts, practitioners, and government to capture Tajikistan's unique primary health care context and to ensure appropriateness for different levels of Tajikistan's health system. The surveys then underwent field testing at selected facilities to further improve the questionnaires based on respondent feedback and identify and address issues that emerge during the pilot. This involved improving option choices for specific questions, improving question framing such as adding clearer and easy to understand definitions of certain terms, and deleting questions that might not be relevant at the facility level.

The final versions of the rapid phone-based and comprehensive in-person HFA questionnaires then underwent a detailed review process to ensure accuracy and consistency in translation. The English versions of both questionnaires were translated to Russian and Tajik by the same translator, reviewed by local context experts, and back translated to English to identify any discrepancies that might affect question clarity.

## Indicator mapping

After adaptation, a systematic indicator alignment process across the rapid phone-based HFA and the comprehensive in-person HFA was conducted. First, all survey items (survey questions) from the seven survey modules of the rapid phone-based HFA (services, infrastructure, human resources, medical supplies and equipment, leadership and coordina-tion, community engagement, quality improvement) were transferred into a Microsoft Excel document. Next, all the survey items from the comprehensive in-person HFA questionnaire that measured the same underlying indicator as the rapid phone-based HFA were identified and mapped to the relevant survey items in the rapid phone-based HFA in the same excel document. At the conclusion of this exercise, 102 survey items from the rapid phone-based HFA were mapped to 129 survey items from the comprehensive in-person HFA. The in-person HFA was therefore deemed 'more comprehen-sive' compared to the rapid phone-based HFA, as the instrument aspires to capture more detail than the phone survey even when restricting the instruments to overlapping domains for the purpose of the validation.

Following the mapping, a question-by-question revision activity was conducted to identify the two types of survey items mapped across both the questionnaires: (1) 'direct' survey items and (2) 'indirect' survey items. Survey items were categorized as 'directly mapped' when the framing and underlying indicator of the survey item were identical (or nearly identical) across both surveys. This required a one-to-one match of a question in the rapid phone-based HFA and the comprehensive in-person HFA. These survey items were commonly the ones wherein the comprehensive in-person HFA question structure was appropriate for phone administration. For instance, both surveys include the same question to identify the source of electricity at health facilities ('What is the health facility's main source of electricity?'). We also considered questions with identical wording across both surveys but which include enumerator verification through direct observation in the comprehensive in-person HFA "directly mapped", because we consider in-person verification as a component of the mode effect. This includes questions on the availability and functioning of facility inputs, such as private toilets, where the enumerators in the comprehensive in-person HFA can verify the response due to being physically present in the health facilities. Directly mapped indicators will be included in objective one (reliability/mode effect) and objective two (validity) testing.

Survey items were classified as 'indirectly mapped' when they intended to measure similar underlying indicators in both surveys, but differed in the way they were asked or framed due to feasibility for phone administration. This typically occurred when comprehensive in-person HFA survey items were too lengthy or detailed to be administered over the phone, leading to simpler alternatives being included in the rapid phone-based HFA. For example, in measuring HIV/AIDS service provision, the rapid phone-based HFA asks whether the health facility provides HIV/AIDS services, while the comprehensive HFA asks whether the health facility was able to provide HIV/AIDS diagnosis services and HIV/AIDS treatment services in the 3 months prior to data collection. In this case, the underlying indicator "HIV/AIDS service availability" was common in both the surveys but was captured differently in each, leading to it being "indirectly mapped."

Since the "indirectly mapped" survey items differed in question framing, they were not suitable for the inter-method reliability test (objective 1) to isolate the mode effect (phone v/s in-person). Any difference in the survey results could be attributed to various factors, such as respondents interpreting the questions differently, rather than solely due to the mode of administration. Therefore, these survey items were suitable only for assessing objective two (validity).

Table 1 provides a summary of the distribution of both types of survey items. An item-by-item mapping is provided in S1 Appendix. The majority (75%) of the questions were deemed 'indirectly mapped' which in and of itself is a finding, demonstrating that few questions are likely to be suitable for identical administration across phone and in-person HFAs.

**Table 1. Number of indicators, their categorization, and relationship to study objectives.**

| | Survey Module | Total number of questions in the study | Indicator Mapping | |
| --- | --- | --- | --- | --- |
| | | | Direct indicators (reliability and validity objectives) | Indirect indicators (validity objective) |
| SERV | Services | 14 | 0 | 14 |
| INF | Infrastructure | 15 | 11 | 4 |
| HR | Human Resources | 8 | 6 | 2 |
| SUP | Medical supplies and equipment | 40 | 0 | 40 |
| LC | Leadership and coordination | 5 | 3 | 2 |
| COM | Communication | 3 | 1 | 2 |
| QI | Quality Improvement | 4 | 4 | 0 |
| **All Areas** | | **89** | **22 (25%)** | **67 (75%)** |

## Data collection

Data collection will take place from August to October 2024. This study has been approved by the Government of the Republic of Tajikistan, with the Order No. 60 and number N2148. Informed consent will be obtained from all survey respondents.

**Sampling.** The sampling methodology for both surveys is designed to be representative of the PHC health facilities in Tajikistan at the national and oblast (region) levels. Since the role of private sector in providing PHC services is minimal in Tajikistan, the sample frame consists of all public PHC health facilities in the country comprising of- City and District Health Centers (family medicine departments), Rural Health Centers, and Health Houses. The sample was stratified across all oblasts. To obtain quality results from the comparison of the rapid phone-based HFA and the comprehensive in-person HFA, each survey will be conducted in the identical sample of 500 facilities described in Table 2.

The enumerators will obtain verbal consent for the rapid phone-based HFA and written consent for the comprehensive in-person HFA from the respondents before starting the survey. If a respondent declines to participate, the reason for their refusal will be documented, and the facility will be replaced by a facility from the same strata whenever available. In the event of a non-response, health facilities will be replaced from a "replacement list" curated from the sample frame for each survey, following the same protocol of random sampling selection. Furthermore, in the event of a non-response, replacements for the two surveys will be done independently. This means that if a facility does not participate in the rapid phone-based HFA and needs to be replaced with another health facility, it will not be simultaneously replaced in the comprehensive in-person HFA, and vice-versa, unless the non-participation is due to the closure of the facility. As a result, the final samples may differ slightly due to varying response rates.

In both surveys, the officer-in-charge of the health facility will serve as the primary respondent. This individual is best positioned to provide comprehensive insights into the facility's overall readiness. However, in situations where the designated officer-in-charge is unable to participate or believes another staff member would be better suited to answer specific sections or questions, the survey response could be delegated to a deputy or another qualified health worker.

**Data collection procedures.** Both surveys will involve comprehensive training for enumerators to familiarize them with survey protocols, questionnaire administration, and ethical considerations for each assessment. This will be followed by additional pilot testing the instrument in a few health facilities to identify any necessary changes and to provide enumerators with field experience before starting data collection.

At least one week before data collection commences, a data collection supervisor will contact the PHC network managers (individuals who manage several public health facilities) via phone to introduce themselves and both surveys. They will also

**Table 2. Sample for the rapid-phone based and comprehensive in-person HFAs.**

|  |  | # Facilities | # Selected |
|---|---|---|---|
| Oblast | Dushanbe | 15 | 15 |
|  | Sughd | 616 | 151 |
|  | Khaktlon | 1,123 | 150 |
|  | Gorno-Badakhshan Autonomous Region | 227 | 84 |
|  | Districts of Republican Subordination | 713 | 100 |
| Type | City Health Center | 41 | 41 |
|  | District Health Center | 53 | 53 |
|  | Rural Health Center | 895 | 214 |
|  | Health House | 1,705 | 192 |
| **TOTAL** |  | **2,694** | **500** |
|  |  |  |  |

request the names and contact information of the relevant sampled health facilities under their supervision. The PHC managers should have received the official notice from the Government regarding the surveys and that their facilities may have been included in the sample. If they do not have the official communication, the supervisor will share with them a copy of the official letter. PHC network managers will then be responsible for notifying officers-in-charge of sampled health facilities.

The rapid phone-based HFA is designed as a short 30–45 minutes survey, requiring no preparation by the officer-in-charge. Respondents will have the option to either start answering the survey during the first call, or to schedule it for a more convenient day or time. If a respondent is unreachable, the rapid-cycle survey protocol requires at least five contact attempts on different days and at varying times to complete the interview, maximizing communication success and reducing potential biases introduced by timing constraints. This will be particularly important for remote facilities with low phone connectivity, where the respondent may have sporadic access to a stable phone network. In instances where immediate completion of the interview is not feasible due to telecommunications issues or respondent availability, follow-up calls may be scheduled for a later date or time.

For the comprehensive in-person HFA, enumerator teams will contact the officers-in-charge of sampled facilities two weeks before data collection, to inform them about the survey, coordinate the date of the first visit, and share the 'advance questionnaire'. The advance questionnaire is a subset of the comprehensive in-person HFA and comprises of questions that require reference to facility data records. This will be shared with the officer-in-charges at least two weeks prior to data collection day, allowing respondents to prepare responses, thereby saving time and improving data quality on the day of data collection.

If facilities have not received the official communication about their participation in the survey, PHC managers will be contacted to ensure that all selected facilities receive the official letter requesting their participation. For the comprehensive in-person HFA, one week before data collection, enumerator teams will again contact the officers-in-charge to confirm the date of the first visit and request completion of the advance questionnaire. This follow-up will also provide an opportunity to address any questions the officers-in-charge may have about the survey and to reschedule the first visit, if necessary. Furthermore, if on the day of data collection, the facility is closed, or the officer-in-charge is not present, or the data cannot be collected for any other reason, the enumerator teams will coordinate and reschedule the visit. If, however, the officers-in-charge refuse to participate in the study, their non-response along with their reason for refusal will be recorded, and the facility will be replaced using random sampling from the required strata.

**Timing of survey administration.** The comprehensive in-person HFA and the rapid phone-based HFA will be conducted as close in timing as possible, ideally within a two-week period for each health facility. The close timing of the surveys minimizes the risk of external factors (for example, a health facility receiving new medical supplies) impacting the data recorded in each survey and will ensure a more reliable comparison. However, there is a risk that conducting both surveys too close will introduce respondent fatigue that might affect results. Therefore, the order of the surveys for each health facility will be randomized to control for some of the real changes that may occur due to the order, with 50% of facilities receiving the rapid phone-based HFA first and 50% of the facilities receiving the comprehensive in-person survey first. This will allow for a more robust analysis. It will also allow us to conduct a longitudinal analysis of responses for the rapid phone-based HFA and the comprehensive in-person HFA in each facility to identify possible confounding effects depending on which survey the facility receives first. We would expect this to be a possibility especially in facilities that receive the in-person survey first, as respondent recall may be primed by the recent in-person verification of survey items and therefore, respondents may answer the rapid phone-based HFA with greater validity.

## Data analysis

Analysis will be conducted on a paired sample of each matched health facility that participated in both surveys.

**Objective one: inter-reliability analysis.** The mode effect (phone v. in-person) of the rapid phone-based HFA compared to the comprehensive in-person HFA will be analyzed using the "directly mapped" survey items. The results

will be analyzed and reported in three ways: as a simple percent agreement of responses, using the Cohen's Kappa to account for the percent of agreement that is expected by chance, and using a prevalence and bias adjusted kappa (PABAK) to account for prevalence and bias in the sample. Calculating PABAK is especially important for a HFA, where many of the survey items are expected to be available and therefore, the expected agreement by chance for many indicators is high due to high prevalence [26].

Taken together, these three measures will provide a nuanced understanding of the results [27].

There are some differences on what is considered adequate for reliability. While literature suggests that a Kappa coefficient larger than 0.61 would be reliable, it is acknowledged that these thresholds can be somewhat arbitrary. To address this, a scale for the level of agreement [28] will be used: almost perfect agreement above 0.90, strong 0.80 to 0.90, moderate 0.60 to 0.79, weak 0.40 to 0.59, and minimal 0.20 to 0.39.

**Objective two: concurrent criterion validity analysis.** This objective focuses on concurrent criterion validity – the extent to which a new measurement technique agrees with a gold standard when the assessments are undertaken at approximately the same time. Both the "directly mapped" and the "indirectly mapped" indicators will be dichotomized to understand the differences in the results between the two surveys with the comprehensive in-person HFA being the gold standard. Sensitivity and specificity will be used to assess whether each indicator meets the predefined validation threshold of 0.7. This has been chosen as the validation threshold because of the specific intent to use low cost, high frequency surveys to generate timely signals as a complement to, rather than replacement for, periodic in-person surveys which are still needed for their depth, comprehensiveness, and rigor. We will also compare results stratified by direct versus indirect indicators and by survey domain (e.g., infrastructure, human resources) to identify any differences.

In addition, regression analysis will be conducted to assess whether sensitivity and specificity are associated with any of the seven domains of the rapid phone-based HFA, facility characteristics, respondent characteristics and potentially enumerator characteristics and timing between the two surveys. Sensitivity checks will also be conducted to see if the duration of time between the two surveys is correlated with the discordant answers.

Regression analysis will also be conducted to identify any variations in the findings that can be attributed to facility characteristics. Subgroup analysis will be conducted based on (1) oblast, (2) rural versus urban, and (3) facility type to identify if any underlying, facility characteristics influence the reliability and validity of the rapid, phone-based HFA results. This will identify whether validity is different in particular sub-groups. For example, the rapid, phone-based HFA may have improved validity in smaller facilities where one respondent is likely to be familiar with all the tracer items and could be more challenging in a larger facility with multiple departments.

Finally, we will compare if the reliability and validity of the rapid phone-based HFA changes based on the order of survey administration. For example, recently conducting the in-person survey may improve the consistency of the responses in the phone survey such that respondents have been primed to respond based on their recent verification of the survey items.

**Objective three: non-response rate.** Although both surveys will initially include the same sample of 500 facilities, we expect that the final samples for each HFA will differ in the end due to varying non-response rates. A simple-t test will be used to assess whether there is a significant difference in the response rates between the rapid phone-based HFA and the comprehensive in-person HFA. Balance tests will also be used to assess whether the characteristics of the final sample of included facilities (the original sample plus any replacements) for the comprehensive in-person HFA and the rapid phone-based HFA sample remain similar across both surveys. Finally, we will document any issues with phone service coverage and the ownership of mobile phones by facilities and respondents, as well as any reported instances of costs to the recipient.

In case a significant difference is found in response rate or sample characteristics, regression analysis will be used to assess associations with discordance in survey completion. This will be done by creating a binary variable to indicate whether the comprehensive in-person HFA was completed in a health facility, while the rapid phone-based HFA wasn't. This binary variable will then regressed on facility characteristics and whether the rapid phone-based HFA was initiated before or after the comprehensive in-person HFA. Since the non-response rate is expected to be higher for the rapid

phone-based HFA, this analysis will only be done for health facilities which complete the comprehensive in-person HFA but not the rapid phone-based HFA.

## Discussion

To the best of our knowledge, this study is the first to assess the reliability and validity of a rapid phone-based HFA against a comprehensive in-person HFA. This will fill an important gap in global evidence on the reliability and validity of phone-based approaches to administer HFAs to monitor a subset of key facility-level indicators. This insight is especially critical given the rapid rise of phone-based survey approaches in LMICs following the COVID-19 pandemic [6] as an innovative method for more frequent, lower-cost, facility-level, primary data collection for select indicators.

### Strengths and limitations

Strengths of this study include careful design of the validation prior to the data collection for both studies, which enabled us to improve analytical rigor by ensuring a common sample, aligned enumerator training processes, and introduction of randomizing in the survey order, and the detailed process undertaken to adapt both surveys to the Tajikistan context. A key limitation is that the final survey instruments for the comprehensive HFA and the rapid phone-based HFA are not identical, comprising both direct and indirect mapped indicators for analytical comparison. We prioritized maintaining as much fidelity to the original phone and in-person instruments as possible, as these two instruments in their original state will be used repeatedly in Tajikistan to monitor large-scale PHC reforms. The challenges we experienced in directly aligning the two instruments is also reflective of our experience that phone and in-person instruments need to be designed and administered differently; therefore, while the lack of identical survey questions is a limitation from the perspective of statistical rigor, it also reflects a practical and operational reality of administering different types of health facility surveys.

Within the study context and the GFF's FASTR initiative more broadly, the intent is to use low cost, high frequency phone surveys as a complement to, rather than a substitute for, periodic and more comprehensive in-person facility assessments. Most immediately, the findings of this study will be used to refine the GFF FASTR Initiative's rapid health facility phone survey assessment prior to its scale-up in multiple countries in Sub-Saharan Africa and Asia in 2025 and beyond. Our intention is that the revised version of the survey instrument and underlying data will then be made publicly available on the GFF's data portal (https://data.gffportal.org/), FASTR's publicly available resource repository (https://data.gffportal.org/key-theme/FASTR/resource-repository/index.php/home), and the SDI Microdata Library (https://microdata.worldbank.org/index.php/catalog/sdi/?page=1&ps=15&repo=sdi). We will also distill our operational learnings in a health facility mobile phone survey manual which will enable the broader public health research and practice community to utilize the FASTR toolset.

More broadly, findings from this study will be used to provide measurement and methodological guidance for HFAs. The findings will provide novel insights on the complementarities of various methods, which HFA domains and sub-domains can be administered via phone with high reliability and validity, and which HFA domains are likely to require in-person administration or other verification approaches. It is possible that the findings could be robust enough to generate correction factors for phone-based surveys that can be applied to approximate in-person measures in contexts where traditional data collection approaches are not possible (such as during health emergencies or conflict-impacted areas). Finally, the findings on response rates will be published to inform future implementation guidance on the feasibility of different methods of data collection given HFA operational constraints and measurement needs in different contexts and when each method is likely to be particularly useful.

## Conclusion

HFAs provide critical information on the quality of health service delivery, making them a vital source of data to inform the design, implementation, and evaluation of health care policies and programs. Validation studies are critical to informing

how and when different types of data collection approaches are appropriate and what can be feasibly measured. This research will improve understanding on the validity of phone-based data collection approaches for HFAs to collect data on a select subset of key service delivery indicators, providing practical insights on the appropriateness, feasibility, and complementarity of different survey modalities in capturing health facility service availability and readiness. This is an important step towards building and sustaining continuous data collection, analysis, and data use for decision making across all levels of the health system in order to improve the quality of service provision, in Tajikistan and globally.

## Supporting information

**S1 Appendix. Appendix Table 1. Indicator mapping comparing the comprehensive in-person health facility assessment and the rapid phone-based health facility assessment.**
(DOCX)

**S1 File. Tajikistan Validation_Inclusivity-in-global-research.**
(DOCX)

## Acknowledgments

For their support in this study, the authors wish to thank the Government of the Republic of Tajikistan and the Ministry of Health and Social Protection of the Population of the Republic of Tajikistan (MOHSPP), in particular Dr. Davlatmurod Olimov, Deputy Head of the State Health and Social Protection Supervision Service (Khadamot) under MOHSPP, and Dr. Ziyodullo Idrisov, Lead Specialist of the MOHSPP Directorate for Health Reforms, PHC and International Cooperation, and as well as the members of the Joint Working Group on Adaptation of Tools for Primary Health Care Quality Survey in Tajikistan. Finally, the authors wish to thank Talip Kilic of the World Bank Group for his comments on an earlier draft of the study protocol. The findings, interpretations, and conclusions expressed in this paper are entirely those of the authors and do not necessarily represent the views of the World Bank, its executive directors, and the governments of the countries they represent.

## Author contributions

**Conceptualization:** Rachel Neill, Pablo Amor Fernandez, Ruchika Bhatia, Jigyasa Sharma, Kathryn Andrews, Sven Neelsen, Mutriba Latypova, Tashrik Ahmed, Michael A. Peters, Ashley Sheffel, Gil Shapira.

**Data curation:** Etoile Pinder, Marifat Abdullaev, Firuza Safarova.

**Funding acquisition:** Mirja Channa Sjoblom, Tawab Hashemi, Peter M. Hansen.

**Project administration:** Etoile Pinder, Marifat Abdullaev, Firuza Safarova, Tawab Hashemi, Ghafur Muhsinzoda.

**Supervision:** Peter M. Hansen, Ghafur Muhsinzoda, Gil Shapira.

**Writing – original draft:** Rachel Neill, Pablo Amor Fernandez, Ruchika Bhatia.

**Writing – review & editing:** Rachel Neill, Pablo Amor Fernandez, Ruchika Bhatia, Jigyasa Sharma, Kathryn Andrews, Sven Neelsen, Etoile Pinder, Marifat Abdullaev, Mutriba Latypova, Mirja Channa Sjoblom, Tashrik Ahmed, Michael A. Peters, Ashley Sheffel, Tawab Hashemi, Peter M. Hansen, Ghafur Muhsinzoda, Gil Shapira.

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
