## [Decision Letter · Decision Letter 0]

21 Jan 2025

PONE-D-24-33961Comparison of the results of in-person and mobile phone surveys for a health facility assessment in Tajikistan: a validation study protocolPLOS ONE

Dear Dr. Neill,

Thank you for submitting your manuscript to PLOS ONE. After careful consideration, we feel that it has merit but does not fully meet PLOS ONE’s publication criteria as it currently stands. Therefore, we invite you to submit a revised version of the manuscript that addresses the points raised during the review process.

We look forward to receiving your revised manuscript.

Kind regards,

Daisuke Nagasato

Academic Editor

PLOS ONE

Journal Requirements:

“For their support in this study, the authors wish to thank the Government of the Republic of Tajikistan and the Ministry of Health and Social Protection of the Population of the Republic of Tajikistan (MOHSPP), in particular Dr. Davlatmurod Olimov, Deputy Head of the State Health and Social Protection Supervision Service (Khadamot) under MOHSPP, and Dr. Ziyodullo Idrisov, Lead Specialist of the MOHSPP Directorate for Health Reforms, PHC and International Cooperation, and as well as the members of the Joint Working Group on Adaptation of Tools for Primary Health Care Quality Survey in Tajikistan. Finally, the authors wish to thank Talip Kilic of the World Bank Group for his comments on an earlier draft of the study protocol. 

Funding for this study was received from the Global Financing Facility for Women, Children, and Adolescents and the World Bank Group. The findings, interpretations, and conclusions expressed in this paper are entirely those of the authors and do not necessarily represent the views of the World Bank, its executive directors, and the governments of the countries they represent.”

Reviewers' comments:

Reviewer's Responses to Questions

**Comments to the Author**

1. Does the manuscript provide a valid rationale for the proposed study, with clearly identified and justified research questions?

Reviewer #1: Yes

Reviewer #2: Yes

2. Is the protocol technically sound and planned in a manner that will lead to a meaningful outcome and allow testing the stated hypotheses?

Reviewer #1: Yes

Reviewer #2: Yes

3. Is the methodology feasible and described in sufficient detail to allow the work to be replicable?

Reviewer #1: Yes

Reviewer #2: Yes

4. Have the authors described where all data underlying the findings will be made available when the study is complete?

Reviewer #1: No

Reviewer #2: No

5. Is the manuscript presented in an intelligible fashion and written in standard English?

Reviewer #1: Yes

Reviewer #2: Yes

6. Review Comments to the Author

You may also provide optional suggestions and comments to authors that they might find helpful in planning their study.

Reviewer #1: The authors describe the protocol for a study to assess the reliability and validity of a rapid phone-based Health Facility Assessment (HFA) in Tajikistan via a comparison against a comprehensive in-person HFA (here used as the reference standard). This is of interest because phone surveys entail cost-savings and could be done more frequently, producing more timely data.

The literature review is comprehensive and the protocol is well described. Below I have listed a few (minor) comments that may improve the manuscript:

(1) Describe the plans for data storage and the modalities for (future) public access of the data.

(2) P6: the evidence linking interview duration to data quality is mixed, e.g., https://doi.org/10.1177/1525822X241237042

(3) I wonder whether informed consent in required given that this is not human subjects research in the strict sense. Respondents report on a public service and not on their own attributes or behavior.

(4) P20 what are ‘heterogeneity effects due to facility characteristics’ ?

(5) P20, L408: correct typo the “predefined” validation threshold …

Reviewer #2: The authors have written. A clear protocol to understand the potential validity of phone facility assessments versus traditional in person assessments. This is an important area of work as understanding and improving care delivery must include more rapid and census level facility assessments which is not feasible given the cost and time of in person facility assessments. The methods need a bit more clarification and overall, the term “phone-based HFA against a comprehensive in-person HFA.” Is a bit misleading as this is not comparing phone based versus comprehensive but phone be versus a subset of questions in one of 3 components of the comprehensive HFA. For example, understand quality (in SDI via vignettes) could be done by phone, and discussion how the 1 component was chosen is needed and adding as a limitation. It would have also been helpful if there was any exploration whether the shorter number of questions give similar results o strength of the readiness in different areas (supply, HR etc) which would help in monitoring change after interventions to address gaps. Some more details below

Introduction’

While it is true that these rapid and more targeted assessment are important in fragile and conflict settings, they are critical to the broader effort in strengthening health systems and the quality of care delivered. They note the 2 HF phone surveys but need to explain what 70% sensitivity for the study in Malawi means. The remainder of the challenges with phone surveys is well detailed (although is there evidence that lower nutritional intake was due to fatigue or potential if there was understanding of the impact of the survey (more food). It is also not clear why a phone survey needs to be only with one person versus more than one call for a facility or having it as a group call

When they look for objective 3, it would also be important to look at phone coverage which in some areas mya also be a barrier as well as cost to the recipient

Methods

The rapid survey should be included in an appendix. A table also mapping where the survey measures versus the HFA

The plan to randomize order is well designed

It would be helpful to understand why some questions were altered (like able to provide versus provide HIV services)

If the OIC decides they are not able to answer a question and delegate it to someone else-is this possible in the phone survey?

Curious why there is advance sharing of a questionnaire for the in person but not the phone survey?

In the analysis-there is no specific plan to analyze differences in responses based on order of administration which is an important methodology plan.

7. PLOS authors have the option to publish the peer review history of their article (what does this mean? ). If published, this will include your full peer review and any attached files.

**Do you want your identity to be public for this peer review?** For information about this choice, including consent withdrawal, please see our Privacy Policy .

Reviewer #1: No

Reviewer #2: No

---

## [Author Response · Author response to Decision Letter 1]

6 Mar 2025

Dear editor and reviewers, thank you for your time in reviewing our manuscript. We have attached a letter and point by point response to this submission which details how we have incorporated your feedback. thank you.

---

## [Decision Letter · Decision Letter 1]

17 Mar 2025

Comparison of the results of in-person and mobile phone surveys for a health facility assessment in Tajikistan: a validation study protocol

PONE-D-24-33961R1

Dear Dr. Neill,

We’re pleased to inform you that your manuscript has been judged scientifically suitable for publication and will be formally accepted for publication once it meets all outstanding technical requirements.

Kind regards,

Daisuke Nagasato

Academic Editor

PLOS ONE

Additional Editor Comments (optional):

Reviewers' comments:

Reviewer's Responses to Questions

**Comments to the Author**

1. Does the manuscript provide a valid rationale for the proposed study, with clearly identified and justified research questions?

Reviewer #2: Yes

2. Is the protocol technically sound and planned in a manner that will lead to a meaningful outcome and allow testing the stated hypotheses?

Reviewer #2: Yes

3. Is the methodology feasible and described in sufficient detail to allow the work to be replicable?

Reviewer #2: Yes

4. Have the authors described where all data underlying the findings will be made available when the study is complete?

Reviewer #2: Yes

5. Is the manuscript presented in an intelligible fashion and written in standard English?

Reviewer #2: Yes

6. Review Comments to the Author

You may also provide optional suggestions and comments to authors that they might find helpful in planning their study.

Reviewer #2: THe autors have done a through job in responding to reviewer comments and the resulting manuscript is strong

7. PLOS authors have the option to publish the peer review history of their article (what does this mean? ). If published, this will include your full peer review and any attached files.

**Do you want your identity to be public for this peer review?** For information about this choice, including consent withdrawal, please see our Privacy Policy .

Reviewer #2: No

---

## [Editor Report · Acceptance letter]

PONE-D-24-33961R1

PLOS ONE

Dear Dr. Neill,

I'm pleased to inform you that your manuscript has been deemed suitable for publication in PLOS ONE. Congratulations! Your manuscript is now being handed over to our production team.

Kind regards,

on behalf of

Dr. Daisuke Nagasato

Academic Editor

PLOS ONE